# Aβ25-35-induced autophagy and apoptosis are prevented by the CRMP2-derived peptide ST2-104 (R9-CBD3) via a CaMKKβ/AMPK/mTOR signaling hub

**Yingshi Ji[1◉], Jinghong Ren[1◉], Yuan Qian[2], Jiaxin Li[1], Huanyu Liu[1], Yuan Yao[1], Jianfeng Sun[3], Rajesh Khanna[4,5]\*, Li Sun[6]\***

1 Department of Pharmacology, College of Basic Medical Sciences, Jilin University, Changchun, Jilin, PR China, 2 Beijing Jishuitan Hospital, Peking University Fourth School of Clinical Medicine, Beijing, PR China, 3 Department of Physiology, Jilin University, Changchun, Jilin, PR China, 4 Department of Pharmacology & Therapeutics, College of Medicine, University of Florida, Gainesville, Florida, United States of America, 5 Pain and Addiction Therapeutics (PATH) Collaboratory, College of Medicine, University of Florida, Gainesville, Florida, United States of America, 6 Department of Neurology and Neuroscience Center, The First Hospital, Jilin University, Changchun, Jilin, PR China

◉ These authors contributed equally to this work.
\* r.khanna@ufl.edu (RK); sunli99@jlu.edu.cn (LS)

**Data Availability Statement:** All relevant data are within the manuscript and its Supporting information files.

## Abstract

We previously reported that the peptide ST2-104 (CBD3, for Ca$^{2+}$ channel-binding domain 3), derived from the collapsin response mediator protein 2 (CRMP2)–a cytosolic phosphoprotein, protects neuroblastoma cells against β-amyloid (Aβ) peptide-mediated toxicity through engagement of a phosphorylated CRMP2/NMDAR pathway. Abnormal aggregation of Aβ peptides (e.g., Aβ25-35) leads to programmed cell death (apoptosis) as well autophagy–both of which contribute to Alzheimer's disease (AD) progression. Here, we asked if ST2-104 affects apoptosis and autophagy in SH-SY5Y neuroblastoma challenged with the toxic Aβ25-35 peptide and subsequently mapped the downstream signaling pathways involved. ST2-104 protected SH-SY5Y cells from death following Aβ25-35 peptide challenge by reducing apoptosis and autophagy as well as limiting excessive calcium entry. Cytotoxicity of SHY-SY5Y cells challenged with Aβ25-35 peptide was blunted by ST2-104. The autophagy activator Rapamycin blunted the anti-apoptotic activity of ST2-104. ST2-104 reversed Aβ25-35-induced apoptosis via inhibiting Ca$^{2+}$/CaM-dependent protein kinase kinase β (CaMKKβ)-mediated autophagy, which was partly enhanced by STO-609 (an inhibitor of CaMKKβ). ST2-104 attenuated neuronal apoptosis by inhibiting autophagy through a CaMKKβ/AMPK/mTOR signaling hub. These findings identify a mechanism whereby, in the face of Aβ25-35, the concerted actions of ST2-104 leads to a reduction in intracellular calcium overload and inhibition of the CaMKKβ/AMPK/mTOR pathway resulting in attenuation of autophagy and cellular apoptosis. These findings define a mechanistic framework for how ST2-104 transduces "outside" (calcium channels) to "inside" signaling (CaMKKβ/AMPK/mTOR) to confer neuroprotection in AD.

**Funding:** This study was supported by the grant provided by the Major Chronic Disease Program of the Ministry of Science and Technology of China (No. 2018YFC1312301), the General Program of the National Natural Science Foundation of China (No. 82071442), the Jilin Provincial Department of Finance (JLSWSRCZX2021-004), the Jilin Science and Technology Department (20240101246JC). All funds were received by Professor Ji Yingshi. The funders had no role in study design, data collection and analysis, decision to publish, or preparation of the manuscript.

**Competing interests:** The authors have declared that no competing interests exist.

## Introduction

Alzheimer's disease (AD is widely regarded as a progressive neurodegenerative disease, clinically characterized by executive dysfunction, declining memory, as well as personality and behavioral changes. The manifold causes include deposition of amyloid-β (Aβ) peptides in extracellular senile plaques, the formation of intracellular neurofibrillary tangles (NFTs), neuroinflammation, and apoptosis [1, 2]. Current research shows that drugs can only delay cognitive impairment in some AD patients but cannot prevent disease progression [3]. The accumulation of Aβ peptide fragments induces neuronal apoptosis and ensuing neurotoxicity [4]. Therefore, the development of drugs that inhibit neuronal death is important to delay the progression of AD.

Studies have shown that neuronal apoptosis is a pathological feature of AD [5]. In recent years, accumulating evidence suggests the occurrence of apoptosis in cultured AD model neurons, animal models of AD, and AD patients [6]. Apoptosis (programmed cell death), a critical physiological process, has an important effect on various cellular processes including growth and function of multicellular organisms [7]. Apoptosis is a key event in the pathobiology of ischemic damage, cancers, and neurodegenerative diseases, including AD [8]. It has been reported that Aβ induces neuronal apoptosis and subsequently triggers AD although the molecular and cellular mechanisms that remain to be deciphered [9]. One mechanism involves Aβ triggering calcium influx, leading to an increase in phosphorylation of tau and generation of reactive oxygen species [10].

Several factors regulate apoptosis, including autophagy. Autophagy involves a series of events during which (i) cytoplasmic proteins are trapped into double-membrane vesicles called autophagosomes, (ii) fuse with lysosomes to produce single-membraned autophagolysosomes, and (iii) are degraded by lysosomal hydrolases [11]. Autophagic cell death is accompanied by the formation of autophagic vacuoles, vacuolation of the endoplasmic reticulum, and moderate chromatin condensation [12].

That $Ca^{2+}$ overload caused by pathological levels of the excitatory neurotransmitter glutamate following ischemic events results in neuronal cell death, a phenomenon referred to as excitotoxicity [13]. The serine/threonine-specific calcium/calmodulin-dependent protein kinase kinase (CaMKK), α and β isoforms, are activated by high intracellular calcium. As a result, downstream targets of CaMKK including Calcium/calmodulin-dependent protein kinase I, isoform I (CaMKI), CaMKIV and AMP-activated protein kinase (AMPK) can be phosphorylated by activated CaMKK [14]. The activated AMPK directly phosphorylates Ser317 or Ser777 of Unc-51 Like Autophagy Activating Kinase 1 (ULK1) to initiate autophagic processes. Mammalian target of rapamycin (mTOR) maintains nutrient utilization via sensing ATP and amino acid levels in the growth of cells. AMPK-ULK1 complex can be suppressed by increased mTOR activity which may further blunt autophagy [15]. Sun and co-workers found that in neurons challenged with oxygen glucose deprivation/reperfusion (OGD/R), propofol regulates a $Ca^{2+}$/CaMKKβ/AMPK/mTOR signaling platform that contributes to regulation and inhibition of autophagy [16].

Serine/threonine-specific calcium/calmodulin-dependent protein kinase kinase (CaMKKβ), which is regulated by cytosolic free calcium $[Ca^{2+}]_c$, acts as an upstream kinase of AMP-activated protein kinase (AMPKα) and regulates important neurological functions, such as synthesis and release of neurotransmitters, ion channels activity, synaptic plasticity and gene expression [17, 18]. It has been reported that Aβ activates CaMKKβ and AMPK in AD primary mouse cortical neurons [19]. Exogenous Aβ increases influx of $Ca^{2+}$ into cells resulting in activation of CaMKKβ and subsequently of AMPK. The activated AMPK directly phosphorylates Ser317 or Ser777 of Unc-51 Like Autophagy Activating Kinase 1 (ULK1) to initiate

autophagic processes [20]. AMPK promotes autophagy by inhibiting the activity of mammalian target of rapamycin (mTOR). Autophagy can cause neuronal apoptosis and aggravate brain damage, illuminating the critical role of the CaMKKβ/AMPK/mTOR signaling hub in mediating neuron death in AD [21].

In recent years, biologics, such as peptides, have gained traction for prevention and amelioration of various diseases like hepatitis, diabetes, and neurodegenerative diseases, primarily due to low toxicity effects. We have advanced small peptides that uncouple protein-protein interactions for excitotoxicity and chronic pain [22–29]. We previously identified CBD3 (for calcium channel binding region; sequence ARSRLAELRGVPRGL), a peptide from the collapsin response mediator protein 2 (CRMP2), which when fused to cell-penetrant motif reversed pain-like behaviors in rodent models [30–32]. CBS3, designated herein as ST2-104, also attenuated nerve damage in brain trauma and cerebral hemorrhage by inhibiting Ca$^{2+}$ influx [33, 34]. In the context of AD, we demonstrated that ST2-104 peptide improved the learning and memory and spatial exploration ability of AD rats.

In this study, we set out to investigate if ST2-104 affects apoptosis and autophagy in SH-SY5Y neuroblastoma challenged with the toxic Aβ$_{25-35}$ peptide (to model AD in vitro) and subsequently mapped the downstream signaling pathways involved. We find that ST2-104 interferes with the increase in Aβ-activated Ca$^{2+}$ influx, attenuates autophagy via affecting the CaMKKβ-AMPK-mTOR signaling hub to affect neuronal apoptosis. Thus, our work has elucidated the molecular mechanism by which ST2-104 inhibits neuron apoptosis in AD.

## Materials and methods

### Cell culture and drug treatment

SH-SY5Y neuroblastoma cells, derived from the SK-N-SH neuroblastoma cells (Key Gen Biotech Co. Ltd. Jiangsu, China), were grown in RPMI 1640 media (Gibco) supplemented with 10% fetal bovine serum (FBS, Biological Industries 04-001-1ACS, Israel) at 37°C with 5% CO$_2$. The SH-SY5Y cell line has been described to produce both substrate (S-type) adherent and neuroblastic (N-type) cells that can undergo trans-differentiation. Therefore, even though the SH-SY5Y cell line is derived from triple successive subclone selection of N-type cells, it contains a small proportion of S-type cells. These cells were plated in 6-well plates (density: $5 \times 10^5$ cells per dish) until confluency reached to ~50%. Cells were then challenged with 5 μM Aβ$_{25-35}$ (Sigma-Aldrich) and 5 to 40 μM ST2-104 peptide (added as a pre-treatment 30 min (Yaoqiang Biological Company, PR China)) and left for 24 h. In separate experiments, cells were pretreated with the mTOR inhibitor (500 nM Rapamycin; Selleck Chemicals) for 4h or the CaMKKβ inhibitor (5 μM STO-609; Selleck Chemicals) for 1h in RPMI 1640 medium.

### Preparation of Aβ$_{25-35}$

One percent acetic acid was used to dissolve Aβ$_{25-35}$ (GSNKGAIIGLM; A4559, Sigma-Aldrich) peptide, which was then diluted in PBS to 1 mM, and subsequently incubated at 37°C for 7 days until Aβ$_{25-35}$ was aggregated. At this stage, the aggregated Aβ$_{25-35}$ was aliquoted and stored at -20°C until use.

### Cell viability assay

SH-SY5Y cells were seeded into 96-well plates (3000 cells/well) and then Aβ$_{25-35}$ (5 μM) was added to trigger cytotoxicity in the presence of acetic acid (control) or ST2-104 peptide (5, 10, 20, 40 μM; this was added 30 min prior to addition of Aβ$_{25-35}$) for 24 h. Next, to each well 5 mg/ml 3-[4,5-dimethylthiazole-2-yl]-2,5-diphenyltetrazolium bromide (MTT) (Sigma-

Aldrich, St. Louis, USA) solution (15 μL/well; Sigma- Aldrich) was added and incubated for 37°C for 4 h. DMSO (150 μl/well) was added for 10 min to dissolve the purple crystals; MTT is a yellow tetrazolium dye that turns purple when it is reduced to an insoluble formazan with DMSO. Finally, the optical density values were examined using a microplate reader (Tecan, Switzerland) at 490 nm. The viability of the control group was defined as 100%.

## Hoechst 33258 staining

To assess for apoptotic cells, morphology of nuclear chromatin was tested using Hoechst 33258 staining (Beyotime, Shanghai, China). Briefly, 4% paraformaldehyde (*vol/vol*) was used to fix cells and then the stained cells were stained with Hoechst 33258 at room temperature for 10 minutes. After washing with PBS, a fluorescence microscope (Olympus, Japan) was employed to visualize nuclear morphology. Apoptotic cells exhibit a condensed nuclear morphology that results from chromatin fragmentation and appears more bright that non-apoptotic cells.

## Intracellular Ca$^{2+}$ measurements

Intracellular calcium was assessed with the fluorescent dye Fluo-3-acetoxymethyl ester probe (Fluo-3AM) (Beyotime, Shanghai, China). Cells were cultured in 24-well plates and treated with Fluo-3 AM (5μM) at 37°C for 1h. Following this incubation, the cells were washed thrice with h PBS. Finally, a flow cytometer (Bio-Rad) was used to excite the cells at 488 nm and monitor their emission of fluorescence at 525 nm, and the gate (M1) was set after collection of the cells. Mean fluorescent intensity (MFI)–a measure of Intracellular calcium–was calculated by Flow Jo.

## Monodansylcadaverine (MDC) staining for autophagosome formation

Morphological changes accompanying autophagy were tested by fluorescence microscopy using monodansylcadaverine (MDC; Solarbio, Beijing, China)) staining. MDC is a fluorescent marker commonly used to stain autophagosomes. Following the indicated treatments of cells on coverslips, cells were rinsed with PBS for 3 times and incubated in 200 μl of 0.05 mM MDC at 37°C for 30 min. Then, the cells were washed thrice with PBS and autophagic vacuole aggregation was observed under a fluorescence microscope with an excitation wavelength of 460–500 nm and an emission wavelength of 512–542 nm.

## Western blot analysis

RIPA buffer (Beyotime) was used to lyse cells treated with various experimental conditions and the lysates were incubated on ice for 30 min. Following centrifugation, protein samples were quantified by bicinchoninic acid protein assay kit (Solarbio). The proteins (~30 μg) were separated on 10%-15% SDS-PAGE gels and then transferred the samples onto PVDF membranes (Bio-Rad). Following blocking in 5% BSA (w/v) or 5% non-fat milk (w/v) for 2 h, the membranes were incubated overnight at 4°C with following validated primary antibodies β-actin (1:10000; Sigma-Aldrich), Bcl-2, Bax, caspase-3, Beclin-1, mTOR, p-AMPK, AMPK (1:1000; Cell Signaling Technology), p-mTOR, LC3-II, CaMKKβ (1:1000; Abcam). Next, the blots were incubated with appropriate secondary antibodies (anti-mouse or anti-rabbit IgG, 1:1000; Beyotime) for 1 h. Between antibody incubations, membranes were washed in TBST three times. The protein bands were detected by the ECL reagent (Bio-Rad) and gray values were analyzed by image J software.

## Statistics

One-way analysis of variance (ANOVA), followed by Tukey post-hoc test, was used for statistical analysis. All data are shown as the mean ± S.E.M. GraphPad Prism software (Version 8.0) was used to perform data analysis. The statistical significance was set at a probability value of P<0.05.

## Results

### R9-CBD3, a CRMP2-derived peptide (i.e., ST2-104), reduces Aβ$_{25-35}$-induced cytotoxicity in SH-SY5Y cells

We first tested if Aβ$_{25-35}$ could induce cytotoxicity by incubating SH-SY5Y cells with 0.5, 1, 5, or 10 μM Aβ$_{25-35}$ for 24 h and 48 h and then assessed cell viability using the MTT assay. As expected, Aβ$_{25-35}$ caused a statistically significant reduction in cell viability at all concentrations tested across both time points with maximal reduction of ~40% at a concentration of 10 μM (Fig 1A and 1B). As shown in Fig 1C, treatment with ST2-104 (0–40 μM, 24 h) alone did not significantly affect cell viability compared with the control treatment. Finally, to ascertain the cytoprotective effect of ST2-104, SH-SY5Y cells were challenged with 5 μM Aβ$_{25-35}$ for 24 h along with increasing concentrations of ST2-104 peptide (5–40 μM) for 24 h. As shown in Fig 1D, all concentrations of ST2-104 peptide increased cell viability, with the 5 μM concentration appearing to be most effective. Although we are uncertain about the exact cause for the lack of observed increase in neuroprotection with the rise in ST2-104 concentration, our hypothesis is that it could be a result of stoichiometric saturation occurring at the minimal concentration of 5 μM; thus we selected 5 μM of ST2-104 peptide for subsequent experiments.

### ST2-104 reduces Aβ$_{25-35}$-induced apoptosis in SH-SY5Y cells

Next, we assessed the cytoprotective effects of ST2-104 on nuclear morphology using Hoechst 33258 staining. As shown in the micrographs in Fig 2A. SH-SY5Y cells treated with 5 μM Aβ$_{25-35}$ has substantially higher levels of condensed nuclei compared with the control or ST2-104 alone treatments, while ST2-104 (5 μM) blocked significantly blocked nuclear condensation when co-applied with Aβ$_{25-35}$ (3 μM). Next, we measured the expression of apoptosis-related proteins in Aβ$_{25-35}$-treated SHY-SY5Y cells using Western blot. Apoptosis-related genes B-cell leukemia-2 (Bcl-2), Bcl-xl, and Bcl-2-associated X protein (Bax) have been reported to be critical in apoptosis. In comparison to the Aβ$_{25-35}$-treated group, the levels of Bax and C-caspase-3 were decreased in the group co-treated with Aβ$_{25-35}$ and ST2-104 (Fig 2B). The expression of Bcl-2 was restored to control levels in the group co-treated with Aβ$_{25-35}$ and ST2-104 (Fig 2B). These data suggest that ST2-104 decreases apoptotic induced by Aβ$_{25-35}$ treatment.

### ST2-104 peptide attenuated Aβ$_{25-35}$-induced SH-SY5Y cells autophagy

Monodansylcadaverine (MDC), a marker for autophagic vacuoles, was used to assess autophagy of SH-SHY5Ycells challenge with Aβ$_{25-35}$. Following staining with MDC, cells were visualized under a fluorescence microscope with the autophagic vacuoles seen as green spots primarily distributed in the perinuclei. Compared to the control group, a significant increase in autophagosomes (green signals in MDC staining) were noted in the Aβ$_{25-35}$-treated group; ST2-104 peptide decreased the number of autophagosomes (Fig 3A).

In order to probe the effects of ST2-104 peptide on SH-SY5Y cells autophagy under the Aβ$_{25-35}$ condition, the autophagosomes was observed by the MDC staining method. In Fig 3A, Aβ$_{25-35}$ group exhibited high intensity of green fluorescence intensity, while ST2-104 peptide

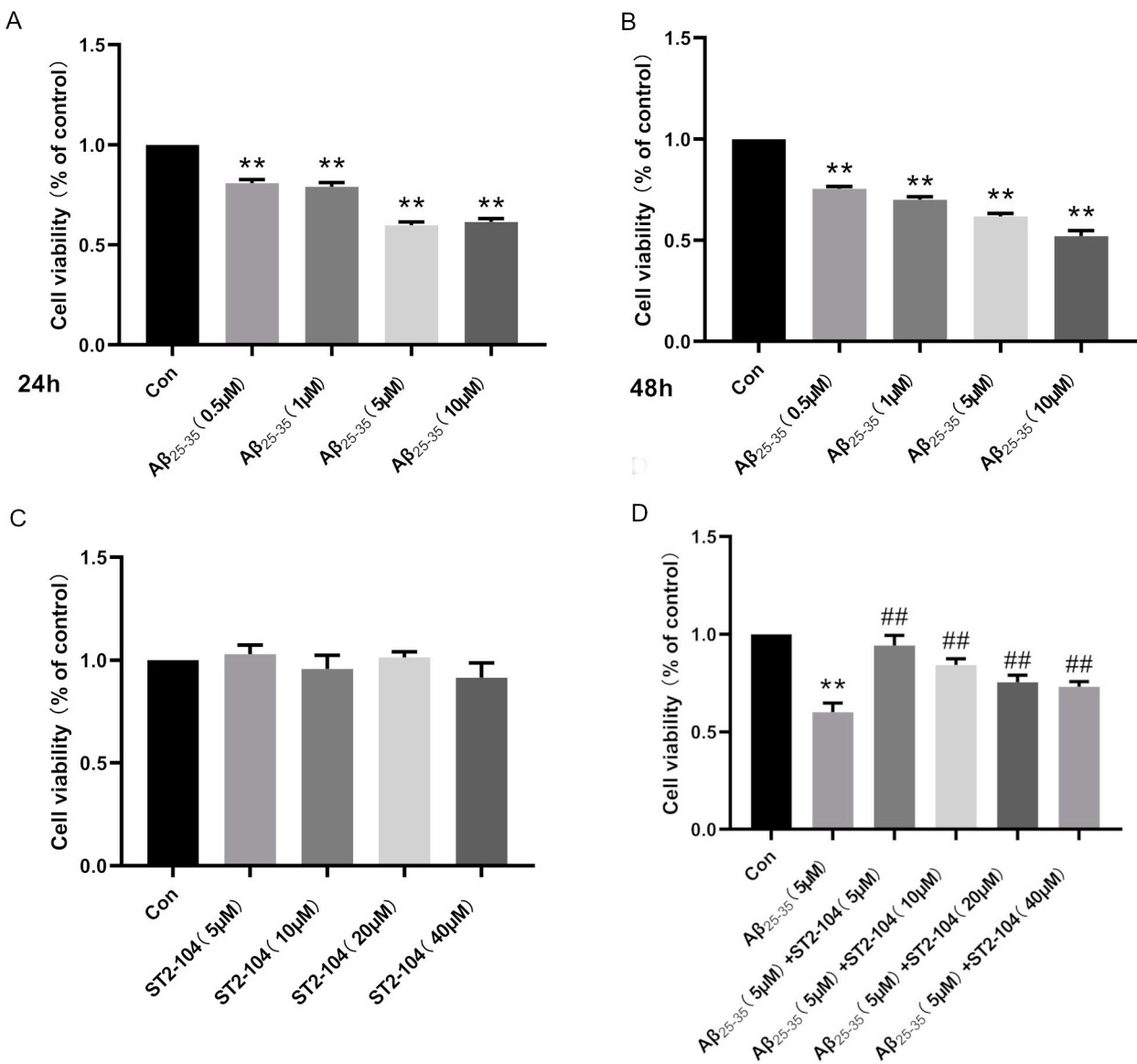

**Fig 1. ST2-104 inhibits SH-SY5Y cell cytotoxicity induced by Aβ$_{25-35}$.** Cell viability of SH-SY5Y cells as assessed by the MTT assay from 6 independent replicates per condition. Conditions were as follows: treatment with 0–10 μM of Aβ$_{25-35}$ treatment for 24 h (A) or 48 h (B); treatment with 0–40 μM of ST2-104 alone for 24 h (C); and treatment with 5 μM Aβ$_{25-35}$ for 24 h along with increasing concentrations of ST2-104 peptide (5–40 μM) for 24 h (D). **$P < 0.01$ vs. control group; ##$P < 0.01$ vs. Aβ$_{25-35}$ group.

significantly decreased the MDC fluorescence intensity under Aβ$_{25-35}$ treatment. Next, we used Western blotting to analyze the level of the autophagy-related proteins LC3-II and Beclin-1. LC3-II is the membrane-bound form of the microtubule-associated protein 1A light chain 3 and serves as a surrogate measure of autophagosome formation while Beclin-1 is an essential mediator of autophagy. Aβ$_{25-35}$ treatment increased the levels of LC3-II and Beclin-1 in comparison to the control group, ST2-104 normalized the levels of these proteins to that observed in control or peptide-alone treated cells (Fig 3B). These data demonstrate that ST2-104 peptide can significantly reduce Aβ$_{25-35}$-induced autophagy of SH-SY5Y cells.

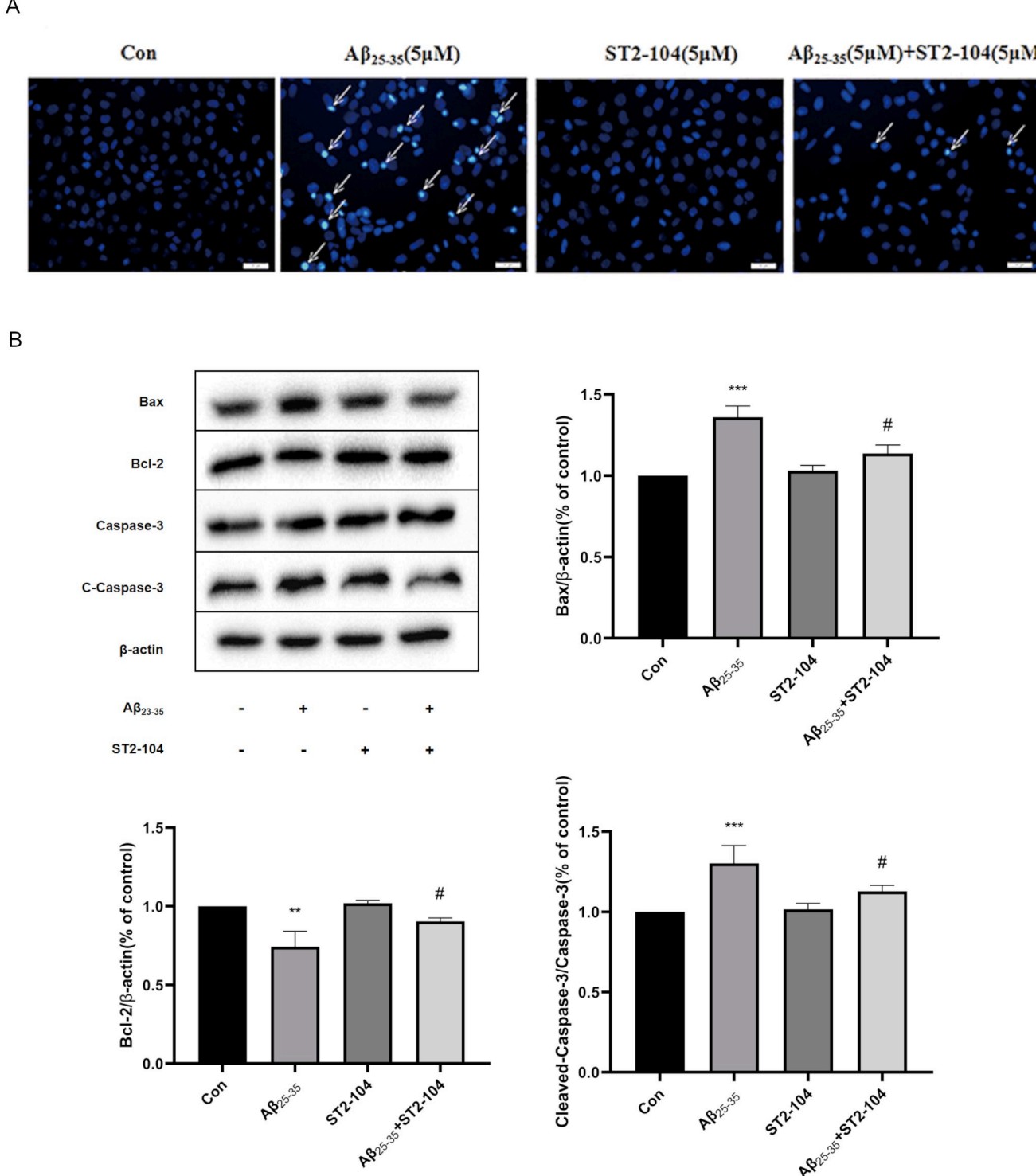

**Fig 2. ST2-104 peptide on SH-SY5Y cells apoptosis induced by Aβ$_{25-35}$.** (A) Apoptosis levels were evaluated using the Hoechst 33258 staining in cells subjected to the indicated treatments. Representative micrographs are shown. Scale bar: 20 μm. (B) Bax, Bcl-2 and C-caspase-3 protein expression were quantified from lysates of SH-SY5Y cells exposed to the indicated treatments for 24 h. Representative western blots are shown as well as the bar graphs of the summary data. *$P<0.05$, **$P<0.01$ vs. control group; #$P<0.05$, ##$P<0.01$ vs. Aβ$_{25-35}$ group. n = 3 per condition.

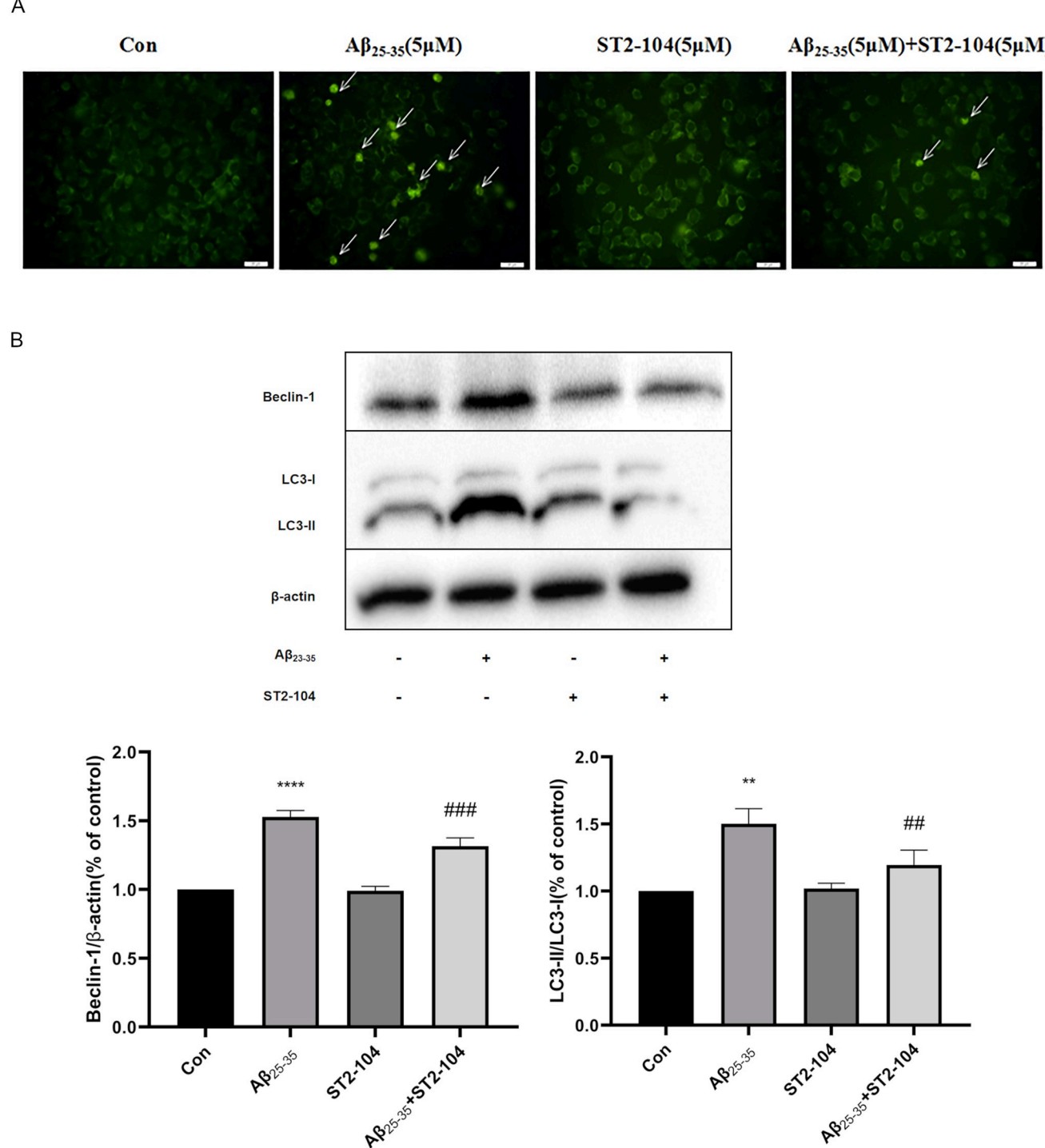

**Fig 3. ST2-104 attenuates Aβ$_{25-35}$-induced autophagy in SH-SY5Y cells.** SH-SHY5Y cells were treated with 5 μM Aβ$_{25-35}$ or control medium or with ST2-104 peptide for 24 h at 37˚C and then autophagy levels and autophagy-related proteins were assessed. (A) Representative images of autophagosomes. Arrowheads indicate autophagosomes marked by MDC staining; scale bar = 20 μm. For each well, at least 5 different fields were examined–a representative is shown here. (B) Detection of LC3-II and Beclin-1 protein levels (representative blot is shown) using Western blot analysis. Levels of β-actin protein were used as the loading control. Bar represents mean ± SEM from 3 separate wells. **P < 0.01 vs. Con group; ##P < 0.01 vs. Glu group with one-way ANOVA with Tukey's post-hoc test.

## Anti-apoptotic activity of ST2-104 peptide is reversed by the autophagy agonist RAPA

To further explore the crosstalk between autophagy and apoptosis in ST2-104 peptide-mediated neuroprotection from $A\beta_{25-35}$, we used ST2-104 peptide combined with the autophagy agonist rapamycin (RAPA) to treat SH-SY5Y cells. The macrolide Rapamycin inhibits the mechanistic target of rapamycin (mTOR) protein kinase. Apoptotic levels were then evaluated with Hoechst 33258 fluorescent staining (Fig 4A). As with our earlier data, $A\beta_{25-35}$ or RAPA alone triggered an increase in apoptosis which was reduced by co-treatment with ST2-104. RAPA negated the reduction in apoptosis conferred by ST2-104. Next, we used western blotting to analyze the expression of the apoptosis-related proteins Bax, C-caspase-3 and Bcl-2. SH-SHY5Y cells treated with $A\beta_{25-35}$ had increased levels of Bax and C-caspase-3 and decreased levels of Bcl-2, these effects were reversed by ST2-104, but normalized by RAPA co-treatment (Fig 4B). Thus, these data show that the autophagy activator RAPA can block the protective effect of ST2-104 peptide on $A\beta_{25-35}$induced apoptosis, suggesting that ST2-104 peptide can block $A\beta_{25-35}$-induced apoptosis by inhibiting apoptosis.

## ST2-104 peptide attenuated SH-SY5Y cells intracellular $Ca^{2+}$ concentration induced by $A\beta_{25-35}$

$A\beta_{25-35}$ treatment has been reported to increase intracellular $Ca^{2+}$ levels. So, we next used the Ca2+-sensitive fluorescence probe Fluo-3/AM to monitor alterations in the intracellular $Ca^{2+}$ by flow cytometry. When SH-SY5Y cells were exposed to 5 μM $A\beta_{25-35}$ for 24 h, the histogram of Fluo-3 fluorescence shifted to a higher intensity (Fig 5), indicating an increase in $[Ca^{2+}]_i$. The mean fluorescence intensity (MFI) and percentage of cells in gate (M1) were changed with $A\beta_{25-35}$-treatment increasing the concentration of intracellular $Ca^{2+}$ (MFI:23.71, M1:25.7%) while co-treatment with ST2-104 peptide (MFI:112.52, M1:10.9%) normalized the $A\beta_{25-35}$-enhanced $Ca^{2+}$ levels to that observed under control conditions (MFI:10.45, M1:7.50%) or cells treated with 5 μM ST2-104 alone (MFI:10.93, M1:8.05%;(Fig 5). These results suggest that ST2-104 can normalize elevated $[Ca^{2+}]_i$.

## Assessing the effects of ST2-104 on the CaMKKβ/AMPK/mTOR signaling hub

Since CaMKKβ activation results in phosphorylated AMPK which can then inhibit the mTOR signaling pathway to trigger autophagy, we asked if expression levels of CaMKKβ/AMPK/mTOR pathway-related proteins could also contribute to $A\beta_{25-35}$ triggered autophagy in SHY-SY5Y cells. The expression levels of CaMKKβ and p-AMPK were enhanced while the level of p-mTOR was reduced by $A\beta_{25-35}$ treatment (Fig 6). ST2-104 significantly decreased the expression levels of CaMKKβ and p-AMPK proteins but increased the level of p-mTOR protein was increased, suggesting that the CaMKKβ/AMPK/mTOR signaling pathway has an important effect on SH-SY5Y cells.

## CaMKKβ contributes to the decrease in apoptosis conferred by ST2-104

It has been reported that a $Ca^{2+}$/CaMKKβ/AMPK/mTOR signaling hub mediates regulation and inhibition of autophagy. To test the involvement of CaMKKβ in ST2-104 peptide-mediated reduction in Aβ-induced apoptosis, the CaMKKβ inhibitor ST0-609 was utilized. ST2-104 decreased glutamate-induced apoptosis and this protective effect was enhanced by STO-609 (Fig 7A). At the protein level, STO-609 decreased the expression of Bax and C-caspase-3 and increased the expression of Bcl-2 significantly compared with glutamate-treated group, similar

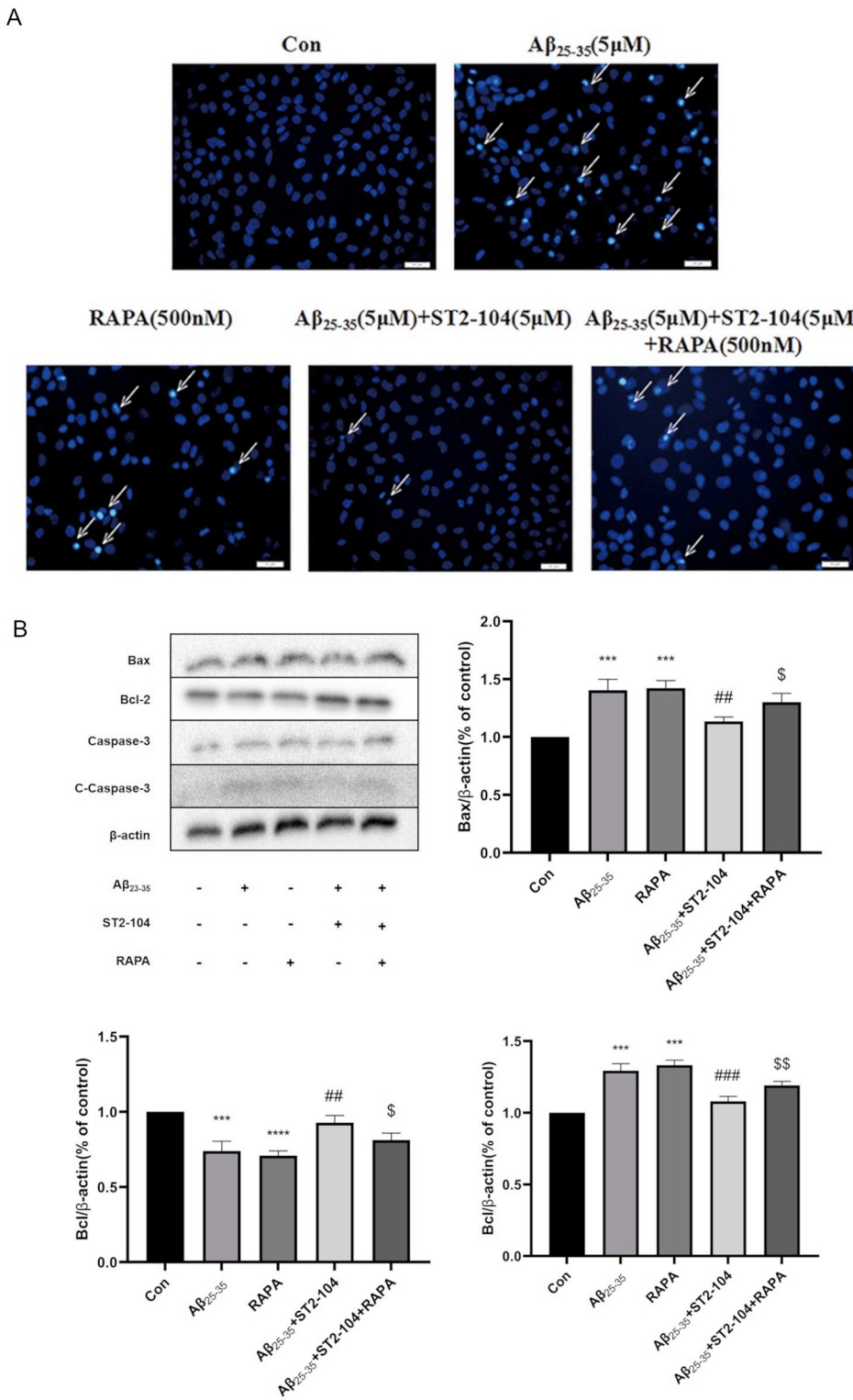

**Fig 4. Reversing ST2-104 peptide-mediated inhibition of autophagy exacerbates apoptotic cell death in SH-SY5Y cells.** (A) Apoptosis of cells treated with the indicated conditions for 24 h was evaluated using the Hoechst 33,258 staining. Scale bar: 20 μm. (B) Detection of Bax, Bcl-2 and C-caspase-3 protein levels (representative blot is shown) using Western blot analysis. Levels of β-actin protein were used as the loading control. $^{**}P<0.01$, vs. control group; $^{##}P<0.01$, vs. Aβ$_{25-35}$ group; $^{\$}P<0.05$, $^{\$\$}P<0.01$, vs. Aβ$_{25-35}$+ST2-104 group (n = 3 per condition).

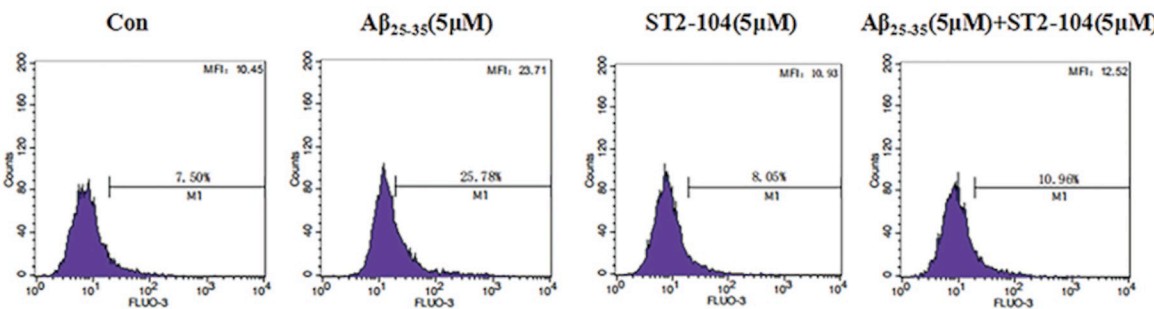

**Fig 5. ST2-104 decreases the Aβ₂₅₋₃₅-mediated increase in intracellular Ca²⁺.** Intracellular calcium ($[Ca^{2+}]_i$) in SH-SY5Y cells was measured by flow cytometry. SH-SHY5Y cells were treated with Aβ₂₅₋₃₅ or control medium or with ST2-104 peptide for 24 h at 37°C and then flow cytometry was performed. $[Ca^{2+}]_i$ was measured by loading the cells with 4 μM of Fluo-3/AM and examining their fluorescence intensity. The results are presented as the mean ± SEM from four independent experiments.

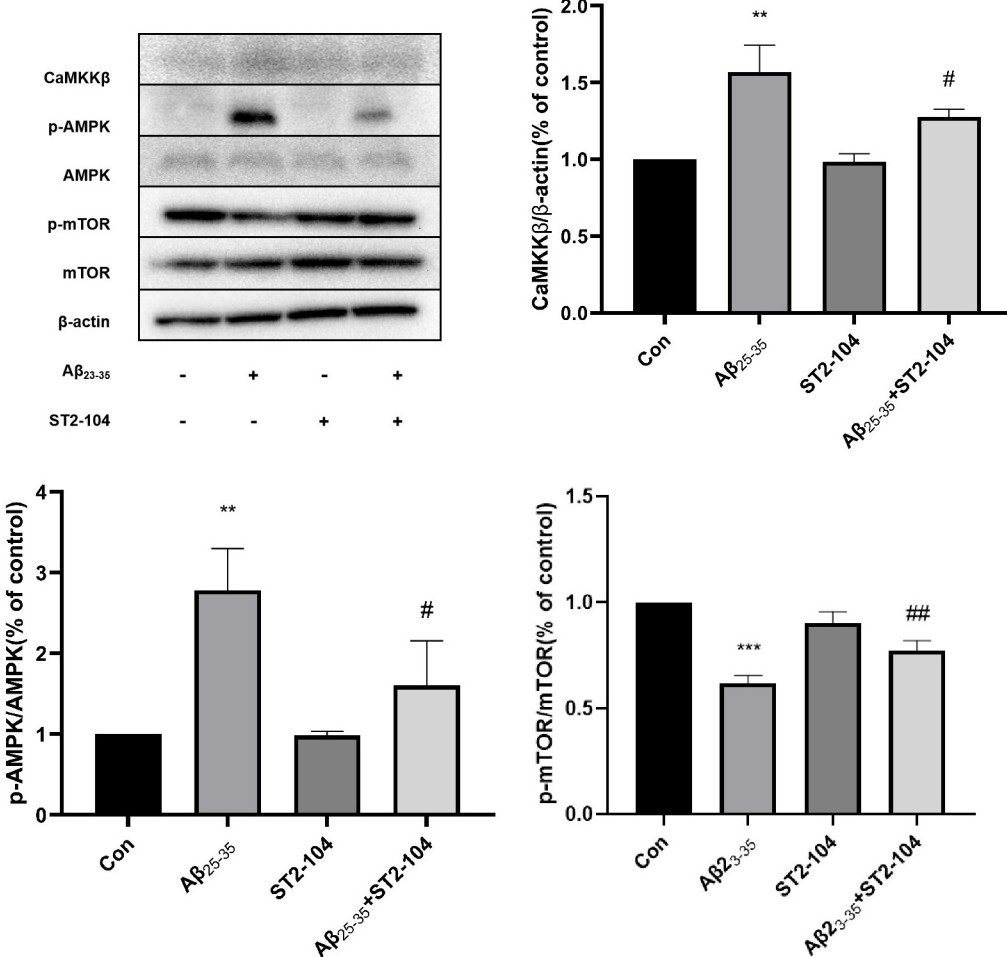

**Fig 6. ST2-104 affects the CaMKKβ/AMPK/mTOR signaling pathway.** SH-SHY5Y cells were treated with Aβ₂₅₋₃₅ or control medium or with ST2-104 peptide for 24 h at 37°C and then protein levels were assessed by Western blotting. Detection of CaMKKβ, AMPK, pAMPK, mTOR and p-mTOR protein expression levels using Western blot analysis. Representative blots are shown. Levels of β-actin protein were used as the loading control. Bar represents mean ± SEM from 3 separate wells. One-way ANOVA with Tukey's post-hoc tests with *$P<0.05$, **$P<0.01$, vs. control group; #$P<0.05$, ##$P<0.01$, vs. Aβ₂₅₋₃₅ group (n = 3 per condition).

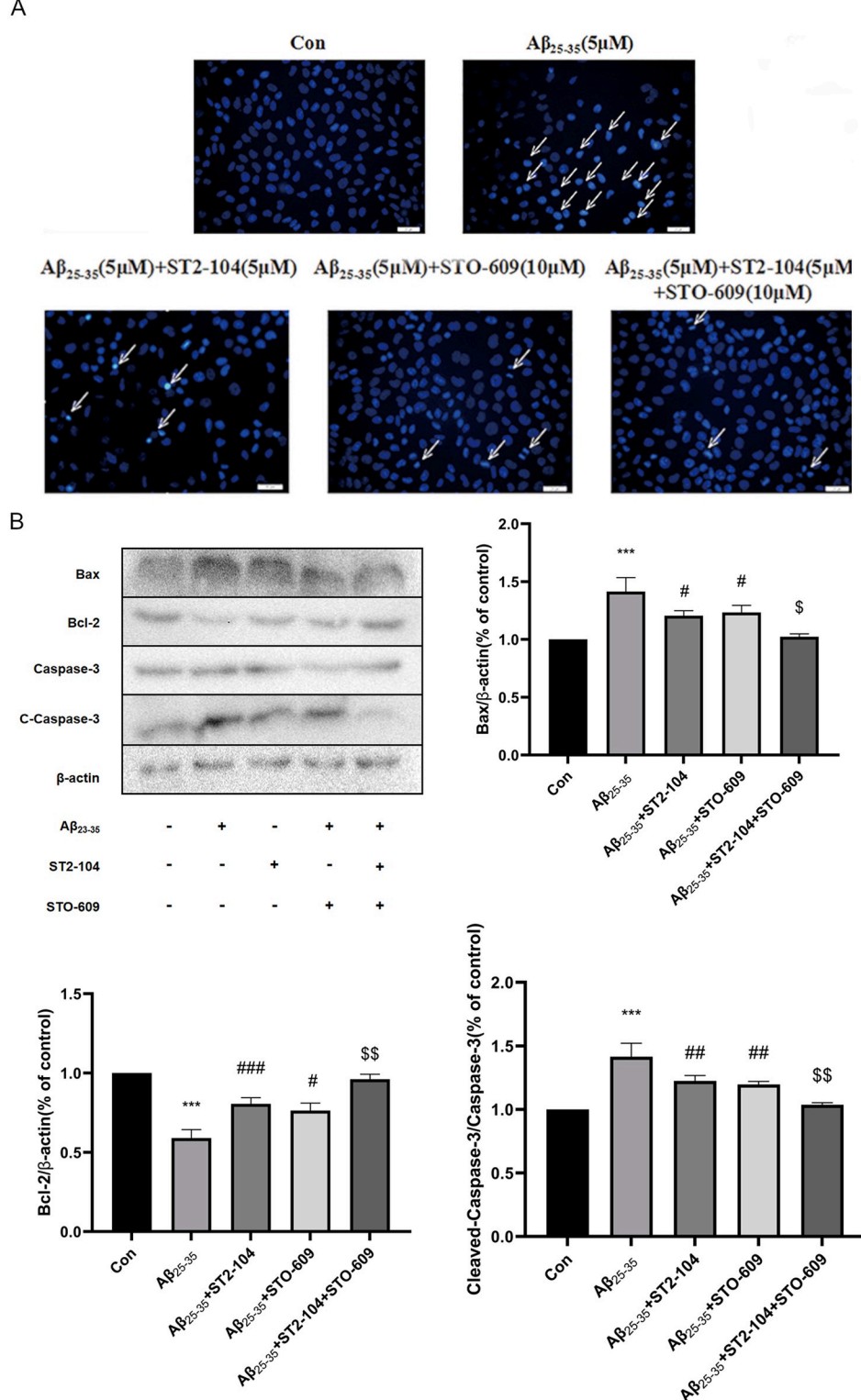

**Fig 7. CaMKKβ contributes to the decrease in apoptosis conferred by ST2-104.** SH-SHY5Y cells were treated with Aβ$_{25-35}$ or control medium or with ST2-104 peptide for 24 h at 37°C and then apoptosis levels and apoptosis-related proteins were assessed. In some wells, 10 μM STO-609, an inhibitor of CaMKKβ was added for 24 h. (A) Apoptosis level was evaluated using the Hoechst 33,258 staining. Scale bar: 20 μm. For each well, at least 5 different fields were examined–a representative is shown here. (B) Detection of Bax, Bcl-2 and C-caspase-3 protein expression levels using Western blot analysis. Representative blots are shown. Levels of β-actin protein were used as the loading control. Bar represents mean ± SEM from 3 separate wells. *$P<0.05$, **$P<0.01$, vs. control group; #$P<0.05$, ##$P<0.01$, vs. Aβ$_{25-35}$ group.

to changes brought about by ST2-104 (Fig 7B). Therefore, ST2-104 peptide inhibits the apoptosis of SH-SY5Y cells by regulating CaMKKβ.

## CaMKKβ is involved in the ST2-104-regulated decrease in autophagy

Finally, we asked if CaMKKβ could affect Aβ$_{25-35}$-induced autophagy. The CaMKKβ inhibitor STO-609 was again used to assay the kinase's role. As shown in Fig 8A, ST2-104 or STO-609 significantly decreased the fluorescence intensity associated with phagolysosomes when compared with cells treated with Aβ$_{25-35}$, as determined MDC staining. Moreover, STO-609 enhanced the fluorescence of the cells which were treated with ST2-104 peptide. At the protein level, ST2-104 or STO-609 decreased the expression levels of Beclin-1, LC3-II, CaMKKβ and p-AMPK, but increased the levels of p-mTOR significantly compared to Aβ$_{25-35}$-treated cells (Fig 8B). Therefore, it appears that that ST2-104 can attenuate autophagy induced by Aβ$_{25-35}$ through the CaMKKβ/AMPK/mTOR pathway in SH-SY5Y cells.

## Discussion

The ST2-104 peptide represents a novel class of interventional strategies. It harbors a Ca$^{2+}$ channel binding domain (CBD3) from a regulatory cytosolic protein (CRMP2) [30]. When added to cell penetrant motifs (e.g., nine arginines), it can uncouple critical interactions between CRMP2 and membrane proteins such as the N-methyl-D-aspartate receptors (NMDARs) [31]. Previous studies found that ST2-104 could reduce Aβ-induced spatial cognitive and memory damage in a rat model of AD, and inhibit Ca$^{2+}$ channel receptor activation, thereby reducing calcium overload, and mitigation of neuronal damage [35, 36]. Based on this evidence, we further explored the role of ST2-104 peptide in neuronal apoptosis induced by the toxic fragment Aβ$_{25-35}$. We found that 5 μM of Aβ$_{25-35}$ for 24 h diminished cell viability to ~60% of control untreated cells while morphological assessment of cells with Hoechst 33258 staining showed high nuclear condensation induced by Aβ$_{25-35}$. At the protein level, the expression levels of Bax, C-caspase-3 were enhanced while Bcl-2 was decreased, supporting the argument that the treatment of 5 μM Aβ25–35 drove apoptosis of SH-SY5Y cells. All of these events triggered by Aβ$_{25-35}$ were ameliorated by ST2-104 pretreatment, establishing an anti-apoptotic role for ST2-104.

Autophagy has also been reported to be dysregulated in AD animal models and AD patients. For instance, immature autophagic vesicles have been observed in dystrophic neurites in brains of AD patients [37]. In ApoE4 transgenic mice, Aβ42 was reported to be significantly higher in lysosomes, eventually contributing to the death of hippocampal neurons [38]. Here, we found that cells treated with Aβ$_{25-35}$ showed higher levels of autophagosomes as revealed by MDC staining. ST2-104 treatment decreased autophagosomes formation. Regarding proteins involved in autophagy, Beclin-1 and LC3-II are key regulators. Beclin-1 interacts with the E3 ligase adaptor autophagy and Beclin-1 regulator 1 (Ambra1), Bif-1, and the tumor suppressor gene UVRAG in the early stage of autophagosome formation to enable activation of the lipid kinase Vps34 (a Class III phosphoinositide 3-kinase), and also interacts with Vps34 to form a core complex which ultimately plays a role in recruiting autophagy related proteins [39]. LC3 is involved in the formation of autophagosomal membranes. With the participation of autophagy related gene 7 (Atg7) and Atg3, cytoplasmic type I LC3 is coupled with phosphatidylethanolamine, and after being modified by ubiquitination, a small piece of polypeptide is enzymatically digested, and then combined with autophagosome membrane to form membrane type II LC3. This transformation is a key step in the formation of autophagosomes. Thus, LC3-II/-I or LC3-II are usually correlated with autophagyous activity [40]. We found that the expression levels of Beclin-1 and LC3-II/LC3-I increased with Aβ$_{25-35}$ treatment. In

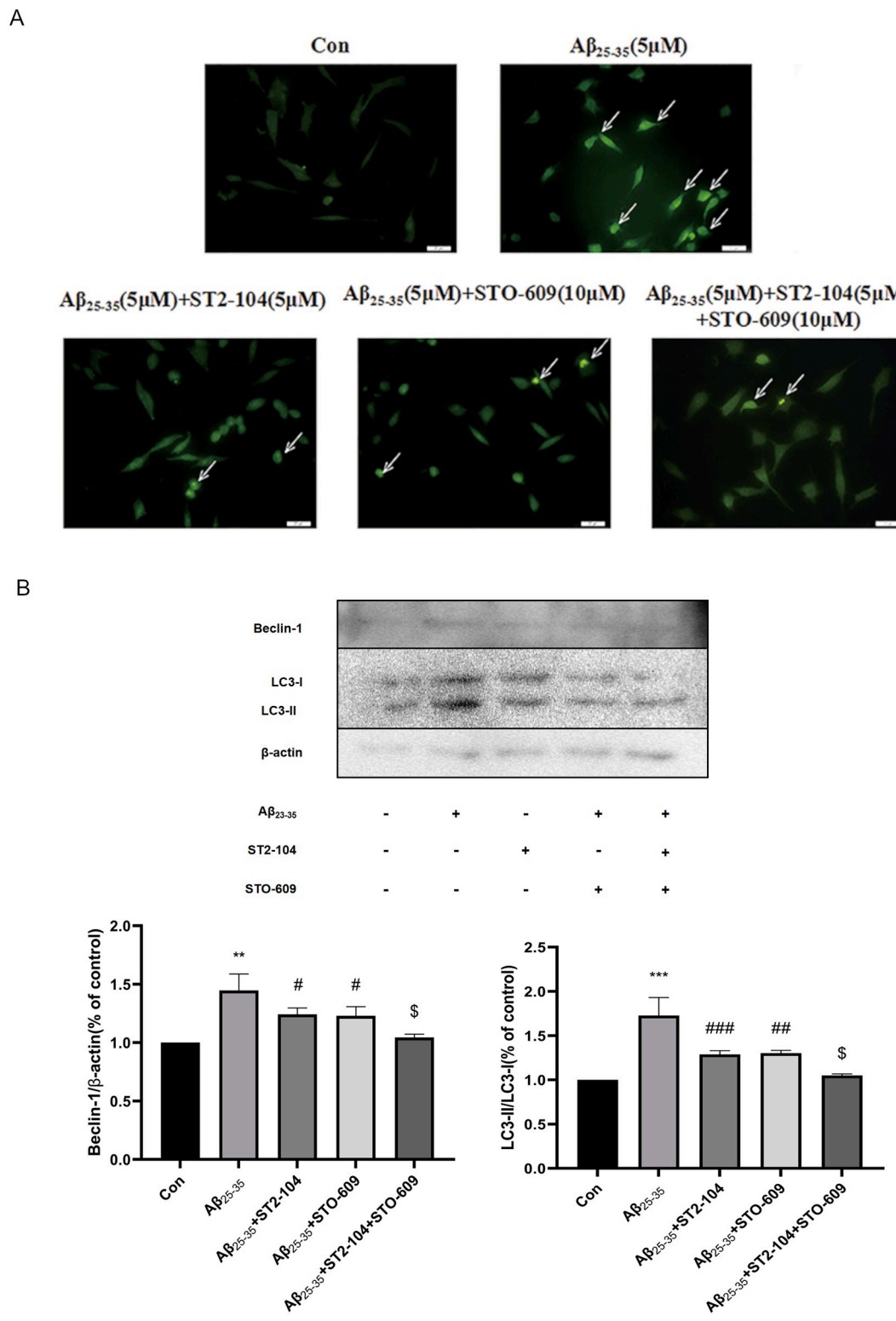

**Fig 8. CaMKKβ is involved in ST2-104-regulated decrease in autophagy.** SH-SHY5Y cells were treated with Aβ$_{25-35}$ or control medium or with ST2-104 for 24 h at 37˚C and then autophagy levels and autophagy-related proteins were assessed. In some wells, 10 μM STO-609, an inhibitor of CaMKKβ was added for 24 h. (A) Autophagy levels were evaluated using MDC staining. Scale bar: 20 μm. For each well, at least 5 different fields were examined–a representative is shown here. (B) Detection of BEclin-1, LC1 and LC3 protein expression levels using Western blot analysis. Representative blots are shown. Levels of β-actin protein were used as the loading control. Bar represents mean ± SEM from 3 separate wells. *$P$<0.05, **$P$<0.01, vs. control group; #$P$<0.05, ##$P$<0.01, vs. Aβ$_{25-35}$ group.

contrast, ST2-104 decreased the expression levels of Beclin-1 and LC3-II with a commensurate reduction in autophagy assessed morphologically. Therefore, our data supports an anti-autophagic role for ST2-104 in prevention of Aβ$_{25-35}$ induced autophagy.

There is increasing evidence that autophagy may have a double-edged effect on brain homeostasis. On the one hand, it is crucial for clearing accumulated misfolded/unfolded proteins and defective organelles [41, 42]. In this case, autophagy is important for maintenance of homeostasis and the suppression of the accumulation of intracellular proteins to toxic levels. However, if the efficiency of lysosomal removal is low, autophagosomes can accumulate in the cell, and the amyloid protein may then be processed into a toxic form. Excessive or dysregulated autophagy activity may promote the production of intracellular Aβ and apoptosis of neurons, which is considered to be a contributory factor in AD pathogenesis [43]. To further explore the interaction between autophagy and apoptosis, we pretreated SH-SY5Y cells with the autophagy agonist rapamycin (RAPA). We found that when autophagy was induced with RAPA, the nuclear condensation and the expression of Bax, C-caspase-3, which is related to enhanced apoptosis, the expression of Bcl-2 was decreased, suggesting RAPA antagonized the protective effect of ST2-104 on SH-SY5Y cells undergoing apoptosis induced by Aβ$_{25-35}$.

Calcium is involved in various neurophysiological activities and is an important signaling ion. As a second messenger in the cell, it participates in numerous activities and is thus kept in strict control. Calcium overload caused by reactive oxygen species and amyloid protein during AD is an important cause of apoptosis. It has been reported that Aβ in AD can affect the steady state levels of $Ca^{2+}$, triggering neuronal excitotoxicity and inducing apoptosis [44]. Our previous research results demonstrated that Aβ fragments can induce activation of glutamate receptor ion channels NMDARs, triggering an increase in $Ca^{2+}$ influx, leading to calcium overload, and ultimately causing synaptic damage [36]. The results of this study showed that Aβ$_{25-35}$ can increase intracellular $Ca^{2+}$ concentration but that ST2-104 decreases the $Ca^{2+}$ influx. Therefore, ST2-104 can inhibit Aβ$_{25-35}$-induced $Ca^{2+}$ accumulation in cells.

To better understand the mechanism of Aβ$_{25-35}$ induced autophagy, we examined the $Ca^{2+}$-related autophagy signaling pathway. CaMKKβ is a protein hub regulated by the levels of $Ca^{2+}$ and one of the upstream kinases that affects AMPK. Studies have found that increased $Ca^{2+}$ can activate CaMKKβ, which is regulated by $Ca^{2+}$ concentration [45]. AMPK, the key factor in energy regulation, can control metabolism by influencing the ratio of AMP to ATP. AMPK negatively regulates mTOR to enhance autophagy and is generally considered to be important in the pathogenesis of AD and other diseases. Studies have also reported that $[Ca^{2+}]_i$ can increase the activation of the CaMKKβ/AMPK signaling, inhibit mTOR signaling, and induce autophagy [46]. We confirmed that when cells are treated with Aβ$_{25-35}$, the levels of CaMKKβ and p-AMPK are enhanced in SH-SY5Y cells while the expression of p-mTOR is decreased. These effects are nullified by ST2-104 pretreatment. Recent studies have shown that the neurotoxicity of Aβ may involve activation of the $Ca^{2+}$/CaMKK/AMPK pathway [21, 47]. Pretreatment of cells with the CaMKKβ inhibitor STO-609 can significantly inhibit AMPK phosphorylation and reduce $Ca^{2+}$-induced autophagy [48]. So, when we used the CaMKKβ inhibitor STO-609, like ST2-104, the level of apoptosis and autophagy were decreased and the levels of CaMKKβ and p-AMPK decreased while that p-mTOR enhanced. Moreover, STO-609 facilitated the beneficial functions of ST2-104, suggesting that ST2-104 down-regulates the autophagy and apoptosis levels of SH-SY5Y cells treated with Aβ$_{25-35}$ to culminate in inhibition of the $Ca^{2+}$/CaMKKβ/AMPK/mTOR signaling axis.

## Conclusion

In conclusion, our results showed that in SH-SY5Y cells, apoptosis induced by Aβ$_{25-35}$ can be inhibited by ST2-104. The mechanism likely involves a sequalae in which ST2-104 inhibits a rise in influx of intracellular calcium induced by Aβ$_{25-35}$, and by doing so affects the CaMKKβ/AMPK/mTOR signaling pathway to curb autophagy and further attenuate apoptosis.

## Supporting information

**S1 Raw images.**
(PDF)

## Author Contributions

**Conceptualization:** Yingshi Ji, Jinghong Ren, Yuan Qian, Jiaxin Li, Huanyu Liu, Yuan Yao.

**Formal analysis:** Jinghong Ren, Yuan Qian, Jiaxin Li, Huanyu Liu, Yuan Yao, Jianfeng Sun.

**Funding acquisition:** Rajesh Khanna, Li Sun.

**Project administration:** Yingshi Ji, Jianfeng Sun.

**Supervision:** Yingshi Ji, Li Sun.

**Validation:** Jinghong Ren, Yuan Qian, Huanyu Liu.

**Visualization:** Yuan Qian, Jiaxin Li, Yuan Yao.

**Writing – original draft:** Yingshi Ji, Jinghong Ren.

**Writing – review & editing:** Yingshi Ji, Jinghong Ren, Yuan Qian, Jiaxin Li, Huanyu Liu, Yuan Yao, Jianfeng Sun, Rajesh Khanna, Li Sun.

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
