## [Decision Letter · Decision Letter 0]

7 May 2024

PONE-D-23-36719Aβ25-35-induced autophagy and apoptosis are inhibited by the CRMP2-derived peptide ST2-104 (R9-CBD3) via a CaMKKβ/AMPK/mTOR signaling hubPLOS ONE

Dear Dr. Sun,

Thank you for submitting your manuscript to PLOS ONE. After careful consideration, we feel that it has merit but does not fully meet PLOS ONE’s publication criteria as it currently stands. Therefore, we invite you to submit a revised version of the manuscript that addresses the points raised during the review process.

We look forward to receiving your revised manuscript.

Kind regards,

Qi Chen, PhD

Academic Editor

PLOS ONE

 [This study was supported by the grant provided by the Major Chronic Disease Program of the Ministry of Science and Technology of China (No. 2018YFC1312301), the General Program of the National Natural Science Foundation of China (No. 82071442), the Jilin Provincial Department of Finance（JLSWSRCZX2021-004）.].  

3. We note that your Data Availability Statement is currently as follows: [All relevant data are within the manuscript and its Supporting Information files]

**Additional Editor Comments:**

Fig 1D, as the concentration of ST2-104 increased from 5 uM to 40 uM, the protective effects was not increasing. Instead, it seemed decreasing. The authors should discuss why.

Fig 3. Autophagy flux should be examined with a liposome inhibitor such as chloroquine.

Fig 7 and Fig 8 lacking a STO-609 control.

Reviewers' comments:

Reviewer's Responses to Questions

**Comments to the Author**

1. Is the manuscript technically sound, and do the data support the conclusions?

Reviewer #1: Yes

2. Has the statistical analysis been performed appropriately and rigorously? 

Reviewer #1: Yes

3. Have the authors made all data underlying the findings in their manuscript fully available?

Reviewer #1: No

4. Is the manuscript presented in an intelligible fashion and written in standard English?

Reviewer #1: Yes

5. Review Comments to the Author

Reviewer #1: The authors have evaluated the protective mechanism of ST2-104 in Aβ25-35 peptide induced cytotoxicity in SHY-SY5Y cells. They showed that ST2-104 prevented the neuronal apoptosis by inhibiting autophagy through a CaMKKβ/AMPK/mTOR signaling pathway. Study of the mechanism of Aβ neurotoxicity is invaluable. Overall, there are several concerns regarding the methodology and interpretation of results.

Major Comments:

Title: Aβ25-35-induced autophagy and apoptosis are inhibited by the CRMP2-derived peptide ST2-104 (R9-CBD3) via a CaMKKβ/AMPK/mTOR signaling hub

Comment: The title should accurately describe the contents of the manuscript. Since the authors used the ST2-104 peptide as a pre-treatment, Aβ25-35-induced autophagy and apoptosis are prevented not inhibited .............

Comments (Methodology and results):

Line 133 &139: What was the solvent of Aβ25-35? DMSO or acetic acid?

Control group received no vehicle? These should be addressed in the method section.

In Line 182, the authors mentioned the use of One-way analysis of variance. The authors should provide the F values in the result section.

The authors have stated in line 280, the percentage of cells in gate (M1). They should provide the flow cytometry protocol of gating and quantification, background noise detection, and an unstained sample.

Comments (Discussion):

In the line 349-356 of Discussion,: The sentences are suitable for introduction not discussion.

The SH-SY5Y cells have two phenotypes “N” and “S” types (Encinas et al. J Neurochem 75:991–1003, 2000). These should be discussed in relation to the current results.

In Line 389, the authors stated that “our data supports an anti-autophagic role for ST2-104 in the face of Aβ25-35 induced autophagy. Since the authors performed the pre-treatment, it is true to write ST2-104 prevented Aβ25-35 induced autophagy.

Please omit “we” in Line 399: "We found that when we autophagy was induced with RAPA, the nuclear condensation and the expression of Bax, C-caspase-"

6. PLOS authors have the option to publish the peer review history of their article (what does this mean?). If published, this will include your full peer review and any attached files.

Reviewer #1: No

---

## [Author Response · Author response to Decision Letter 0]

14 Jun 2024

PONE-D-23-36719

Response to reviewers

Our responses are indicated in red font below.

 [This study was supported by the grant provided by the Major Chronic Disease Program of the Ministry of Science and Technology of China (No. 2018YFC1312301), the General Program of the National Natural Science Foundation of China (No. 82071442), the Jilin Provincial Department of Finance（JLSWSRCZX2021-004）.].  

Please include this amended Role of Funder statement in your cover letter; we will change the online submission form on your behalf. We have added this to the cover letter.

3. We note that your Data Availability Statement is currently as follows: [All relevant data are within the manuscript and its Supporting Information files]

We have provided all the raw data in an accompanying Supplementary file.

Additional Editor Comments:

Fig 1D, as the concentration of ST2-104 increased from 5 uM to 40 uM, the protective effects was not increasing. Instead, it seemed decreasing. The authors should discuss why.

While the exact reason for why we did not observe increasing neuroprotection upon increasing the concentration of ST2-104 is not known, we hypothesize that this may be due to stoichiometric saturation at the lowest concentration of 5 µM. 

Fig 3. Autophagy flux should be examined with a liposome inhibitor such as chloroquine. We indeed plan to measure this in future work. 

Fig 7 and Fig 8 lacking a STO-609 control. Controls for STO-609 have been previously published (see PMIDs: 37710272 and 36889111). While we should have used it as a control in our studies, we acknowledge this as an oversight and limitation of our work here. 

Reviewers' comments:

Reviewer's Responses to Questions

Comments to the Author

1. Is the manuscript technically sound, and do the data support the conclusions?

Reviewer #1: Yes

2. Has the statistical analysis been performed appropriately and rigorously?

Reviewer #1: Yes

3. Have the authors made all data underlying the findings in their manuscript fully available?

Reviewer #1: No

4. Is the manuscript presented in an intelligible fashion and written in standard English?

Reviewer #1: Yes

5. Review Comments to the Author

Reviewer #1: The authors have evaluated the protective mechanism of ST2-104 in Aβ25-35 peptide induced cytotoxicity in SHY-SY5Y cells. They showed that ST2-104 prevented the neuronal apoptosis by inhibiting autophagy through a CaMKKβ/AMPK/mTOR signaling pathway. Study of the mechanism of Aβ neurotoxicity is invaluable. Overall, there are several concerns regarding the methodology and interpretation of results.

Major Comments:

Title: Aβ25-35-induced autophagy and apoptosis are inhibited by the CRMP2-derived peptide ST2-104 (R9-CBD3) via a CaMKKβ/AMPK/mTOR signaling hub

Comment: The title should accurately describe the contents of the manuscript. Since the authors used the ST2-104 peptide as a pre-treatment, Aβ25-35-induced autophagy and apoptosis are prevented not inhibited .............

We have amended the title to reflect the prevention.

Comments (Methodology and results):

Line 133 &139: What was the solvent of Aβ25-35? DMSO or acetic acid?

Control group received no vehicle? These should be addressed in the method section.

The solvent for Aβ25-35 was acetic acid, and an equal volume of acetic acid was added to the control group. we will modify the content in the Methods section.

In Line 182, the authors mentioned the use of One-way analysis of variance. The authors should provide the F values in the result section. F-values for the relevant data are provide in the Supplementary excel file.

The authors have stated in line 280, the percentage of cells in gate (M1). They should provide the flow cytometry protocol of gating and quantification, background noise detection, and an unstained sample. We provide examples of the gating, quantification, noise detection and an unstained sample in a supplementary figure.

Comments (Discussion):

In the line 349-356 of Discussion,: The sentences are suitable for introduction not discussion. Noted and removed.

The SH-SY5Y cells have two phenotypes “N” and “S” types (Encinas et al. J Neurochem 75:991–1003, 2000). These should be discussed in relation to the current results. We now state that “The SH-SY5Y cell line has been described to produce both substrate adherent (S-type) and neuroblastic (N-type) cells that can undergo transdifferentiation. Therefore, even though the SH-SY5Y cell line is derived from triple successive subclone selection of N-type cells, it contains a small proportion of S-type cells.”

Please see the references: 

Forster JI, Köglsberger S, Trefois C, Boyd O, Baumuratov AS, Buck L, Balling R, Antony PM. Characterization of Differentiated SH-SY5Y as Neuronal Screening Model Reveals Increased Oxidative Vulnerability. J Biomol Screen. 2016 Jun;21(5):496-509. doi: 10.1177/1087057115625190. Epub 2016 Jan 6. PMID: 26738520; PMCID: PMC4904349.

In Line 389, the authors stated that “our data supports an anti-autophagic role for ST2-104 in the face of Aβ25-35 induced autophagy. Since the authors performed the pre-treatment, it is true to write ST2-104 prevented Aβ25-35 induced autophagy. We have corrected this.

Please omit “we” in Line 399: "We found that when we autophagy was induced with RAPA, the nuclear condensation and the expression of Bax, C-caspase-" Done.

6. PLOS authors have the option to publish the peer review history of their article (what does this mean?). If published, this will include your full peer review and any attached files.

Do you want your identity to be public for this peer review? For information about this choice, including consent withdrawal, please see our Privacy Policy.

Reviewer #1: No

While revising your submission, please upload your figure files to the Preflight Analysis and Conversion Engine (PACE) digital diagnostic tool, https://pacev2.apexcovantage.com/. PACE helps ensure that figures meet PLOS requirements. To use PACE, you must first register as a user. Registration is free. Then, login and navigate to the UPLOAD tab, where you will find detailed instructions on how to use the tool. If you encounter any issues or have any questions when using PACE, please email PLOS at <a href="mailto:figures@plos.org">figures@plos.org. Please note that Supporting Information files do not need this step.

---

## [Decision Letter · Decision Letter 1]

28 Jun 2024

PONE-D-23-36719R1Aβ25-35-induced autophagy and apoptosis are inhibited by the CRMP2-derived peptide ST2-104 (R9-CBD3) via a CaMKKβ/AMPK/mTOR signaling hubPLOS ONE

Dear Dr. Sun,

Thank you for submitting your manuscript to PLOS ONE. After careful consideration, we feel that it has merit but does not fully meet PLOS ONE’s publication criteria as it currently stands. Therefore, we invite you to submit a revised version of the manuscript that addresses the points raised during the review process.

We look forward to receiving your revised manuscript.

Kind regards,

Qi Chen, PhD

Academic Editor

PLOS ONE

Journal Requirements:

Additional Editor Comments:

Please incorporate changes in response to the previous comments into the appropriate place in the manuscript.

Reviewers' comments:

Reviewer's Responses to Questions

**Comments to the Author**

1. If the authors have adequately addressed your comments raised in a previous round of review and you feel that this manuscript is now acceptable for publication, you may indicate that here to bypass the “Comments to the Author” section, enter your conflict of interest statement in the “Confidential to Editor” section, and submit your "Accept" recommendation.

Reviewer #1: All comments have been addressed

2. Is the manuscript technically sound, and do the data support the conclusions?

Reviewer #1: Yes

3. Has the statistical analysis been performed appropriately and rigorously? 

Reviewer #1: I Don't Know

4. Have the authors made all data underlying the findings in their manuscript fully available?

Reviewer #1: Yes

5. Is the manuscript presented in an intelligible fashion and written in standard English?

Reviewer #1: Yes

6. Review Comments to the Author

Reviewer #1: 1. The revised manuscript was reviewed. The authors have responded to the most of this reviewer's comments. The author’s explanations to queries must be incorporated in appropriate place in the manuscript. However, they referred to supplementary files. For instance, the discussion about the SH-SY5Y cells phenotypes or the F values.

In my opinion, it is ok if the supplementary files are published with the manuscript.

2. The first sentence of discussion” The ST2-104 peptide represents a novel class of such strategies.” What does it mean? which strategies? It should be corrected.

7. PLOS authors have the option to publish the peer review history of their article (what does this mean?). If published, this will include your full peer review and any attached files.

Reviewer #1: No

---

## [Author Response · Author response to Decision Letter 1]

18 Aug 2024

PONE-D-23-36719R1

Response to reviewers

Our responses are indicated in red font below.

Please review your reference list to ensure that it is complete and correct. If you have cited papers that have been retracted, please include the rationale for doing so in the manuscript text, or remove these references and replace them with relevant current references. Any changes to the reference list should be mentioned in the rebuttal letter that accompanies your revised manuscript. If you need to cite a retracted article, indicate the article’s retracted status in the References list and also include a citation and full reference for the retraction notice. We checked the reference list to make sure it is complete and correct.

Additional Editor Comments:

Please incorporate changes in response to the previous comments into the appropriate place in the manuscript.

Reviewers' comments:

Reviewer's Responses to Questions

Comments to the Author

1. If the authors have adequately addressed your comments raised in a previous round of review and you feel that this manuscript is now acceptable for publication, you may indicate that here to bypass the “Comments to the Author” section, enter your conflict of interest statement in the “Confidential to Editor” section, and submit your "Accept" recommendation.

Reviewer #1: All comments have been addressed

2. Is the manuscript technically sound, and do the data support the conclusions?

Reviewer #1: Yes

3. Has the statistical analysis been performed appropriately and rigorously?

Reviewer #1: I Don't Know

4. Have the authors made all data underlying the findings in their manuscript fully available?

Reviewer #1: Yes

5. Is the manuscript presented in an intelligible fashion and written in standard English?

Reviewer #1: Yes

6. Review Comments to the Author

Reviewer #1: 1. The revised manuscript was reviewed. The authors have responded to the most of this reviewer's comments. The author’s explanations to queries must be incorporated in appropriate place in the manuscript. However, they referred to supplementary files. For instance, the discussion about the SH-SY5Y cells phenotypes or the F values.

In my opinion, it is ok if the supplementary files are published with the manuscript.

2. The first sentence of discussion” The ST2-104 peptide represents a novel class of such strategies.” What does it mean? which strategies? It should be corrected. We have changed “such strategies” to “interventional strategies”.

7. PLOS authors have the option to publish the peer review history of their article (what does this mean?). If published, this will include your full peer review and any attached files.

Do you want your identity to be public for this peer review? For information about this choice, including consent withdrawal, please see our Privacy Policy.

Reviewer #1: No

While revising your submission, please upload your figure files to the Preflight Analysis and Conversion Engine (PACE) digital diagnostic tool, https://pacev2.apexcovantage.com/. PACE helps ensure that figures meet PLOS requirements. To use PACE, you must first register as a user. Registration is free. Then, login and navigate to the UPLOAD tab, where you will find detailed instructions on how to use the tool. If you encounter any issues or have any questions when using PACE, please email PLOS at <a href="mailto:figures@plos.org">figures@plos.org. Please note that Supporting Information files do not need this step.

---

## [Editor Report · Decision Letter 2]

20 Aug 2024

Aβ25-35-induced autophagy and apoptosis are prevented by the CRMP2-derived peptide ST2-104 (R9-CBD3) via a CaMKKβ/AMPK/mTOR signaling hub

PONE-D-23-36719R2

Dear Dr. Sun,

We’re pleased to inform you that your manuscript has been judged scientifically suitable for publication and will be formally accepted for publication once it meets all outstanding technical requirements.

Kind regards,

Qi Chen, PhD

Academic Editor

PLOS ONE
---

## [Editor Report · Acceptance letter]

16 Sep 2024

PONE-D-23-36719R2 

PLOS ONE

Dear Dr. Sun, 

I'm pleased to inform you that your manuscript has been deemed suitable for publication in PLOS ONE. Congratulations! Your manuscript is now being handed over to our production team.

Kind regards, 

on behalf of

Dr. Qi Chen 

Academic Editor

PLOS ONE